# An Efficient Plugin Method for Metric Optimization of Black-Box Models

## Abstract

Many machine learning algorithms and classifiers are available only via API queries as a "black-box" — that is, the downstream user has no ability to change, re-train, or fine-tune the model on a particular target distribution. Indeed, a downstream user may not have any knowledge of the training distribution or performance metric used to construct and optimize the black-box model. We propose a simple and efficient method, CWPLUGIN, which takes as input arbitrary multiclass predictions, and post-processes them in order to adapt them to a new target distribution and to optimize a particular metric of the confusion matrix. Importantly, CWPLUGIN is a *post-hoc* method which does not rely on feature information, only requires a small amount of probabilistic predictions along with their corresponding true label, and optimizes metrics by querying. We empirically demonstrate that CWPLUGIN has performance competitive with related methods on a variety of tabular and language tasks.

## 1 Introduction

Consider the following common scenario: A machine learning practitioner would like to adapt a public, open source model to a particular target task with only small set of labeled target examples. There are a plethora of approaches in domain and task adaptation for working in this setting, including model fine-tuning (Han et al., 2024; Dodge et al., 2020), low-rank adaptation (Hu et al., 2022), classical importance weighing techniques (Azizzadenesheli, 2021; Lipton et al., 2018; Sugiyama et al., 2007), and more (see, e.g., Ganin & Lempitsky (2015); Sun & Saenko (2016); You et al. (2019)). These methods have been relatively successful, and show that the underlying base model can be improved or modified in order to adapt its performance to the target distribution quite efficiently.

The modern machine learning landscape, however, has become rife with *proprietary* and *black-box* models. For example, there are numerous image and language APIs which allow for only query access to the models of interest. For example, developers using Google's vision API (Google, 2024), Amazon's Rekognition (Amazon, 2024), or Clarifai's platform (Clarifai, 2024) are usually restricted from accessing or tuning the underlying model, and can only interact with it via API requests. In light of this more challenging setting, we revisit the fundamental question of model adaptation:

> *If a machine learning practitioner has only **black-box query access** to a model, when and how can they adapt the model to a particular target task with only a small number of labeled examples?*

We will assume that the only information which the model designers share is *class probability estimates* for any queried data point — particular details about the training distribution, training loss, model weights, or even the model architecture itself are unknown. In this more restricted setting, most fine-tuning or re-training approaches are immediately disqualified since the underlying model architecture, weights, or training data are all unavailable to the practitioner.

In addition to distribution shift, we also consider how a practitioner can adapt the predictions of a black-box model in order to optimize a specific *metric* of interest other than the one the model was trained to optimize. The cross-entropy loss is the de-facto objective optimized in order to achieve good performance on metrics such as accuracy and calibration; however, at test or production time, system designers may also desire prioritizing other metrics such as F-measures (Ye et al., 2012;

Puthiya Parambath et al., 2014), geometric mean and classifier sensitivity (Monaghan et al., 2021), Matthews Correlation Coefficient (Chicco & Jurman, 2020), and more (Müller et al., 2022). As an example, a practitioner utilizing models for downstream tasks such as sorting patients to receive clinical attention (Hicks et al., 2022) or utilizing a closed-source language model to screen CVs (Gan et al., 2024), the performance of the classifier on a particular metric of interest — e.g., minimizing a particular mix of false-positives and true-positives — may be more important than simply obtaining good accuracy. Indeed, some performance metrics of interest may not even have a closed form, and can only be estimated by deploying a production grade system to a target population (Huang et al., 2021; Hiranandani et al., 2021).

Taken together, methods which can adapt classifiers in a *post-hoc* and *black-box* manner to (1) account for distribution shift; and (2) optimize specific metrics have broad applicability. Both tasks are especially salient given the recent history and potential evolution of the model landscape (Maslej et al., 2024).

**Contributions.**   We propose a simple and effective coordinate-wise plugin method CWPLUGIN for post-processing the probabilistic predictions of a *black-box* predictor in order to simultaneously achieve both (1) improved performance on a *shifted distribution*; and (2) improvement on a specified *metric* of interest. CWPLUGIN method is broadly applicable since it only assumes *query* access to the metric, and is not defined inherently defined by assuming any structure of the metric itself.

We introduce CWPLUGIN in Section 3.1. As input, the algorithm takes in (1) a set of probabilistic multiclass predictions on a target domain along with their true labels; and (2) query access to a particular *metric* of interest (e.g., accuracy, recall, F-measure). We consider metrics which can be defined as simple functions of the confusion matrix, as standard in the black-box classification literature (Hiranandani et al., 2020; Jiang et al., 2020). The output of CWPLUGIN is a set of $m$ class weights, one for each of $m$ classes. These weights are then used at inference time in order to appropriately re-weigh each of the classes in order to maximize the metric of interest.

In Section 3.2, we demonstrate that for a certain class of metrics — linear diagonal metrics — plugin is a *consistent* classifier in that it will eventually recover the Bayes optimal predictor under the metric of interest. We also demonstrate that the design of CWPLUGIN allows for its run-time to be substantially improved when data is class balanced or the metric it is optimizing obeys a certain quasi-concavity property (Section 3.3).

Since the only inputs to CWPLUGIN are raw multiclass predictions — and not feature data — it is an extremely flexible method which can be applied to a variety of both classical and modern domains. To demonstrate this, in Section 4 we provide experimental evidence of its superior performance for metric optimization across multiple tabular and language classification tasks under distribution shift. For an illustrated setting of where CWPLUGIN may be applied, we refer to Figure 1.

## 1.1   RELATED WORK

Classfier metric optimization is a well studied problem in both theory and practice (Ye et al., 2012; Koyejo et al., 2014; Narasimhan et al., 2014; Yan et al., 2018). Most related, however, is the line of work investigating optimizing *black-box metrics*, e.g., when no closed form of the metric is known Zhao et al. (2019); Ren et al. (2018); Huang et al. (2019); Hiranandani et al. (2021). This line of work utilizes a variety of approaches, including importance weighed empirical risk minimization, or model retraining for robustness. The most relevant work is that of Hiranandani et al. (2021), which is a purely post-hoc method which does not require retraining or fine-tuning classifiers. The authors there propose a post-hoc estimator which is learned via a "probing classifier" approach. Their approach solves a particular, global linear system in order to find the weights which optimize a particular metric. Our proposed method is instead *local* in that it considers only pair-wise comparisons between classes. We demonstrate the superior performance of our method on a variety of real-world black-box prediction tasks, suggesting that a global, linear system approach may not always be necessary.

There is a long history of work in machine learning on domain adaptation, or generalization under distribution shift. These fall into a few main categories: Distributionally Robust Optimization (DRO, Rahimian & Mehrotra (2019)), Invariant Risk Minimization (IRM, Arjovsky et al. (2019)), various importance weighing methods (Lipton et al., 2018), and many more (Wilkins-Reeves et al., 2024; Gretton et al., 2008; Nguyen et al., 2010). As far as we are aware, however, there are few

methods other than calibration which operate using only (probabilistic) predictions and labels for the target distribution, and further do not require re-training or fine-tuning of the original model. These properties are essential, as they allow methods to be applied on top of closed-source models (Geng et al., 2024). One such example is the work of Wei et al. (2023), who propose re-weighing predictions in the face of distribution *prior* shift with DRO.

Calibration has long been a staple method within the machine learning community (Platt et al., 1999; Niculescu-Mizil & Caruana, 2005; Guo et al., 2017; Minderer et al., 2021; Carrell et al., 2022). Any probabilistic classification model can be provably calibrated in a post-hoc manner, even for *arbitrarily* distributed data (Gupta et al., 2020). Recently, Wu et al. (2024) demonstrated that a stronger version of calibration from the algorithmic fairness literature, multicalibration (Hébert-Johnson et al., 2018), has deep connections to robustness under distribution shift, and proposed a post-processing algorithm which adapts a predictor under both co-variate and label shift for regression tasks.

It is worth mentioning that language models have their own set of domain adaptation techniques, such as fine-tuning from supervised (Han et al., 2024) or human feedback (Tian et al., 2023), prompt tuning/engineering (Liu et al., 2023), in-context learning (Dong et al., 2022), etc. Our method is agnostic to the choice of underlying base model; nonetheless, we include fine-tuning as a suitable baseline where applicable.

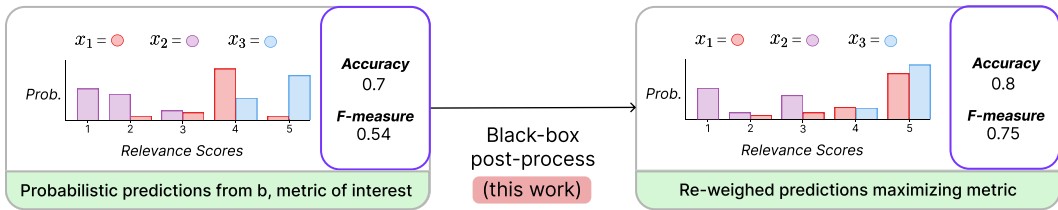

Figure 1: The setting of our work. As input (Left), our method takes arbitrary probabilistic, multiclass predictions (along with true labels) on a target distribution from a black-box model $b$. The bars are conditional label probabilities of data points $x_1, x_2$, and $x_3$, and the x-axis shows classes. A metric of interest (e.g., Accuracy, F-measure, etc.) is also given as input. The CWPLUGIN algorithm then post-processes these predictions in a black-box manner, without any re-training or fine-tuning of the underlying model. The resulting probabilistic predictions (Right) have their performance on the selected metric of interest improved.

## 2 PRELIMINARIES

Let $\mathcal{X}$ be the data domain and $\mathcal{Y} = \{1, 2, \ldots, m\} = [m]$ be the set of labels in a multiclass classification problem. Let $\Delta(\mathcal{Y})$ denote the set of all *distributions* over labels. A (probabilistic) *predictor* $b : \mathcal{X} \to \Delta(\mathcal{Y})$ maps data points to distributions over classes. We call $b$ a *black-box predictor* if we do not have any knowledge of how $b$ was created, its particular architecture, or how it functions. Indeed, we may not even know or have access to the *source* distribution that $b$ was trained on: all we have is *query* access to obtain $b(x)$ for any given $x \in \mathcal{X}$. Typical examples of black-box predictors include closed-source models of classification API services such as Google VisionAI or Amazon Rekognition (Google, 2024; Amazon, 2024), custom text classification solutions provided by a company like Clarifai (Clarifai, 2024), or models trained on proprietary health data and made available to us via API by independent entity (see, e.g., Dandelion (2024)).

We call $S = \{(b(x_i), y_i)\}_{i \in [n]}$ a sample of $n$ data points, and assume that $(x_i, y_i) \sim D$ i.i.d. for a *target* distribution $D$ supported on $\mathcal{X} \times \mathcal{Y}$. Notice that we adopt the convention of using the predictions of $b$ to define the sample $S$; this is purely to simplify notation since our proposed method will operate using only the predictions of $b$ (and disregard any feature information). We work in the scenario where $|S|$ is small, say, on the order of tens or hundreds of examples. Therefore, given that the target and source domains have non-trivial overlap, we expect that training or fine-tuning a *new* model from scratch using only the sample $S$ will give sub-par performance on the target domain.[1]

**Metrics and Confusion Matrices.** Before discussing how we plan to improve $b$ by re-weighing its predictions, we first provide background on the metrics we seek to optimize. As is standard in the black-box classification literature (Hiranandani et al., 2021; Jiang et al., 2020), we consider post-processing $b$ in order to optimize for metrics defined as functions of the *confusion matrix*. We

---

[1]Indeed, we investigate this assumption more rigorously in our experiments.

measure the performance of a (deterministic) classifier $h : \mathcal{X} \to \mathcal{Y}$ on $S$ using the empirical *confusion matrix* $\mathbf{C}^h \in [0,1]^{m \times m}$, which, at entry $\mathbf{C}^h_{i,j}$, measures the fraction of data in $S$ which is of true class $i \in [m]$, but classified by $h$ as $j \in [m]$. We measure the performance of a randomized classifier $g : \mathcal{X} \to \Delta(\mathcal{Y})$ in an identical way—we simply take the prediction of the classifier at input $x$ to be the $\arg\max$ over predicted probabilities.

Many metrics of interest can be captured by *functions* of the confusion matrix $f : \mathbf{C}^h \mapsto \mathbb{R}_{\geq 0}$. For example, accuracy is simply the trace of the confusion matrix: $f_{\mathrm{acc}}(\mathbf{C}^h) = \mathrm{Tr}(\mathbf{C}^h)$, or for a binary classification problem, the F-measure of $h$ can be written as $f_{\text{F-1}}(\mathbf{C}^h) = 2 \cdot \mathbf{C}^h_{1,1}/(2 \cdot \mathbf{C}^h_{1,1} + \mathbf{C}^h_{0,1} + \mathbf{C}^h_{1,0})$. Similar equations can be found for multiclass F-measure, geometric mean, etc (see, e.g., Narasimhan et al. (2023)). Throughout, we adopt the convention that larger values of $f$ are better.

## 3 REWEIGHING PREDICTIONS USING LEARNED CLASS WEIGHTS

In Section 3.1, we propose CWPLUGIN: a method for learning weights $\mathbf{w}$ to re-weigh the predictions from a black-box predictor $b$ in order to optimize a metric $f$, potentially under distribution shift between the source domain that $b$ was trained on and the novel target domain. We argue that CWPLUGIN is simple to implement, and can be analyzed in a certain restricted setting (Section 3.2). We also show that it is generally parallelizable, and with certain additional structure of the metric $f$, enjoys sizable efficiency improvements (Section 3.3).

### 3.1 THE CWPLUGIN RE-WEIGHING METHOD

Our proposed method will learn a vector $\mathbf{w} \in \mathbb{R}^m$ of $m$ weights, one to re-weigh each of the $m$ classes predicted by $b$. Simply re-weighing the predictions is surprisingly expressive: Not only does it allow for provably optimizing certain families of metrics (Section 3.2), it also describes the Bayes optimal learner under certain kinds of *distribution shift* such as label shift and label noise (see, e.g., Hiranandani et al. (2021, Table 1)). In addition, there are a variety of post-hoc model adaptation methods from the calibration and robustness literature which show surprising potential improvements by modifying the output of a predictor $b$ with only $m$ or $m^2$ parameters (Guo et al., 2017; Kull et al., 2019; Wei et al., 2023; Wang, 2023); We use these as motivation in our design of CWPLUGIN.

A naive approach to learning the optimal weights $\mathbf{w}^*$ maximizing the metric $f$ on the sample set $S = \{(b(x_i), y_i)\}_{i \in [n]}$ is to perform a *brute-force* $m$ dimensional grid search over $[0,1]^m$. For binary classification problems, this simplifies to tuning the decision threshold to optimize a metric $f$ using a hold-out validation set.[2] However, this approach quickly becomes infeasible as the number of classes $m$ grows beyond two and the required precision $\epsilon$ increases. For example, finding $\mathbf{w}^*$ with $m = 5$ classes and precision $\epsilon = 0.1$, or $m = 3$ and $\epsilon = 0.01$, both require a search over at least $10^5$ grid points — and hence, metric evaluations — of $f(\mathbf{C}^h)$.

To ameliorate this, we instead propose a *coordinate-wise* search approach, which we call CWPLUGIN. Instead of performing a grid search over all $O(1/\epsilon^m)$ grid points, CWPLUGIN *fixes* one of the classes — say, class $m$ — as a reference class. It then restricts consideration to the $m - 1$ classifiers which output either class $k$ or class $m$ everywhere (for $k \in [m-1]$). It will use these restrictions in order to find an optimal *relative weight* between each pair of classes.

Before formalizing this, we introduce the following necessary assumption on the black-box predictor $b$ in order to guarantee convergence of CWPLUGIN. Let $b(x)_k$ be the probability of class $k \in [m]$.

**Assumption 1.** *For each $k \in [m]$, there exists $x_j \in S$ such that $b(x_j)_k > 0$.*

This assumption simply states that the sample $S$ is non-trivial over all $m$ classes. This is w.l.o.g.: if $b$ did not satisfy this for some class $k$, we could simply drop that class from all predictions.

With this assumption in hand, consider the hypothesis $h^{k,m}_\alpha$ which uses $b$ to either predict only either class $k$ or $m$ on every input, written as:

$$h^{k,m}_\alpha(b(x)) = \begin{cases} k & \text{if} \quad \alpha b(x)_k > (1-\alpha)b(x)_m \\ m & \text{otherwise.} \end{cases} \tag{1}$$

---

[2]See, for example, `TunedThresholdClassifierCV` in scikit-learn (Kramer & Kramer, 2016).

---

**Algorithm 1** CWPLUGIN

---

1: **Input:** Sample $S = \{(b(x_i), y_i)\}_{i \in [n]}$, Number of classes $m$.
2: **Initialize:** $\mathbf{w} = \mathbf{1} \in \mathbb{R}^m$.
3: **for** $k \in [m-1]$ **do** $\qquad\qquad\qquad\qquad\qquad$ ▷ Iterate over each class pair $(k, m)$
4: $\qquad$ Let $S_{k,m} = \{(b(x_j), y_j) \mid y_j \in \{k, m\}\} \subseteq S$ $\qquad$ ▷ Restrict $S$ to samples in class $k$ or $m$
5: $\qquad$ $\alpha_k = \arg\max_{\alpha \in [0,1)} f(\mathbf{C}^{h_\alpha^{k,m}})$ ▷ Find best $\alpha$ for restricted classifier $h_\alpha^{k,m}$ in Equation (1)
6: $\qquad$ Set $\mathbf{w}_k = \alpha_k / (1 - \alpha_k)$. $\qquad\qquad\qquad$ ▷ Set $\mathbf{w}_k$ to best relative weight for class $k$ over $m$
7: **end for**
8: **Set:** $\mathbf{w} = \frac{\mathbf{w}}{\sum_{k=1}^m \mathbf{w}_k}$ $\qquad\qquad\qquad\qquad\qquad$ ▷ Normalize weights to ensure $\mathbf{w} \in [0,1]^m$
9:
10: **Inference**: To classify new, unseen data $x \in \mathcal{X}$, predict $h_{\text{plugin}}^{\mathbf{w}}(x) = \arg\max_{k \in [m]} b(x)_k \mathbf{w}_k$.

---

Notice that $h_\alpha^{k,m}$ is derived from the predictor $b$ by predicting class $k$ or $m$ based on which of $\frac{\alpha}{1-\alpha} b(x)_k$ or $b(x)_m$ is larger. The reason for considering this restricted binary classifier is as follows. The predictor $h_\alpha^{k,m}$ will only ever output class $k$ or class $m$ over the entire sample $S$. This means that by tuning $\alpha \in [0,1]$, we can find the $\alpha = \alpha_k$ which provides the best metric value $f(\mathbf{C}^{h_{\alpha_k}^{k,m}})$ for $h_\alpha^{k,m}$ with the empirical confusion matrix $\mathbf{C}^{h_\alpha^{k,m}}$ constructed with the sample $S$. Given that there exists $x$ such that $b(x)_k > 0$ for any $k$ (Assumption 1), such an $\alpha$ value is guaranteed to exist. This optimization can be done to precision $\epsilon > 0$ with a line search in $O(1/\epsilon)$ time for any pair of classes $(k, m)$. Lastly, we normalize all these relative weights so that the returned $\mathbf{w}$ lies in $[0,1]^m$.

A full description of CWPLUGIN is given in Algorithm 1. After obtaining the weights $\mathbf{w}$, we augment the black-box predictor $b$ by taking the weighted prediction $b_{\mathbf{w}}(x) = \arg\max_{k \in [m]} b(x)_k \mathbf{w}_k$.

**Discussion.** We note that choosing class $m$ to be fixed is arbitrary: this can easily be changed to any other class $k \in [m]$ with little impact to the algorithm (with enough samples). Furthermore, since for each pair $(k, m)$ of classes, Algorithm 1 considers only the restriction of $S$ to $S_{k,m}$ — the data points in $S$ with true class label $k$ or $m$ — the *order* that the algorithm iterates over classes does not impact the final chosen solution. That is, each relative weight $\mathbf{w}_k$ is independent from all others. Finally, we note that it suffices to run the line search in line 5 over $\alpha \in [0, 1-\rho]$ for sufficiently small $\rho > 0$. As we show in the Appendix (Proposition 6), the value of $\rho$ can depend on the metric of interest $f$; in practice, however, we simply take it to be $\rho = \epsilon$, the granularity of our line search.

Intuitively, the restriction to *pair-wise* class relevance scores is inherently local; Algorithm 1 can only evaluate how each class should be weighed relative to one another, then modify the frequency at which the different classes are predicted. Nonetheless, as we show in Section 4, this approach can often provide performance gains with only a few samples.

## 3.2 ANALYSIS

In this section, we sketch the guarantees of the CWPLUGIN algorithm within the framework of *metric weight elicitation* (Zhao et al., 2019). Most details are deferred to Appendix A.2, but we give an informal overview of our results here. The metric weight elicitation framework assumes that the metric $f$ is only available via oracle query. The goal is to *learn* the metric $f$, by assuming that it has a specific functional form (linear, diagonal, etc.), and fitting the relevant coefficients using a sample $S$.

In our first result, Proposition 6, we show that CWPLUGIN is a *consistent* estimator for the family of *linear-diagonal* metrics: it elicits the optimal weights and learns the Bayes optimal predictor when given access to population quantities. Afterwards, in Proposition 9, we show that with a finite (polynomial) number of samples, CWPLUGIN can still obtain approximately optimal weights for the underlying linear-diagonal metric Both results illustrate that CWPLUGIN may provide rigorous statistical guarantees in the presence of metric shift; this is not normally provided by standard post-hoc post-processing methods like calibration.

## 3.3 SPEEDING UP CWPLUGIN

We now study the efficiency of CWPLUGIN, and show that it can be significantly improved with particular types of metrics, class-balanced datasets $S$, or parallelization. To begin with, we first

analyze the runtime of the algorithm as stated in Algorithm 1. This version uses a line search to optimize for $\alpha \in [0, 1 - \epsilon]$ in line 5 of the algorithm.

**Proposition 2.** *The runtime of* CWPLUGIN *in Algorithm 1 is $O(mn/\epsilon)$.*

*Proof.* For each pair of classes $(k, m)$ for $k \in [m - 1]$, we must check the value of the metric $f$ $1/\epsilon$ times[3], once for each possible setting of $\alpha_k \in [0, 1 - \epsilon]$. Assume that running a metric evaluation $f(\mathbf{C}^h)$ on the empirical confusion matrix $\mathbf{C}^h$ of a dataset $S$ of size $n$ requires time $O(n)$. Then, the total runtime of CWPLUGIN with line search is $O(mn \cdot \frac{1}{\epsilon})$. $\qquad\square$

To work towards improving this, we consider a *restricted* class of metrics for which faster run-time is possible via replacing the line search with binary search.

**Lemma 3.** *Let $f$ be a metric such that for all pairs of classes $(k, m)$ for $k \in [m - 1]$, the restricted metric $f(\mathbf{C}^{h_\alpha^{k,m}})$ from line 5 in Algorithm 1 is quasi-concave over the domain $\alpha \in [0, 1 - \epsilon]$. Then, the number of metric evaluations in Algorithm 1 can be improved from $O(m/\epsilon)$ to $O(m \log(1/\epsilon))$. In particular, the line search in line 5 of Algorithm 1 can be improved to a binary search.*

The proof is deferred to Appendix A.1. Perhaps the broadest class of metrics which satisfies this pair-wise quasi-concavity property — beyond simply linear-diagonal metrics — is that of *linear-fractional* diagonal metrics, which can be written as $f(\mathbf{C}^h) = \frac{\langle \boldsymbol{a}, \mathrm{Diag}(\mathbf{C}^h) \rangle + b}{\langle \boldsymbol{b}, \mathrm{Diag}(\mathbf{C}^h) \rangle + d}$ with a strictly positive denominator. This family of metrics can include certain variants of, for example, F-measure and $\beta$ F-measure (Hiranandani et al., 2019b).

A summary of the run-times of Algorithm 1 is available in Table 1. Notice that perfectly class balanced data can remove the dependence on $m$ completely. We also remark that for a cost of $O(n)$ memory overhead, CWPLUGIN can be parallelized to potentially remove up to a factor of $m$ from the stated run-times for "worst-case" $S$ (not necessarily class-balanced). This is because the order of the optimization over classes $k \in [m - 1]$ does not matter, i.e., the **for** loop of lines 3-7 in Algorithm 1 can be *parallelized*. The only shared memory will be the restriction of the sample $S$ to data points of true class $m$. This implies that with only $O(n)$ additional memory, the overall running time may be greatly reduced with multi-threading or parallelization.

|  | Line Search | Binary Search (quasi-concave $f$ only) |
|---|---|---|
| Worst-case $S$ | $O(mn/\epsilon)$ | $O(mn \log(1/\epsilon))$ |
| Class-Balanced $S$ | $O(n/\epsilon)$ | $O(n \log(1/\epsilon))$ |

Table 1: Run-times for Algorithm 1 with various optimizations and class balanced data.

## 4 EXPERIMENTS

We provide preliminary empirical evidence that the CWPLUGIN method can be used post-hoc to improve the metrics of black-box predictors in various distribution shift / metric optimization settings.

**Experimental Setup.** In our experimental setup, we will work with three different sets of data. The *training set* is sampled from the source distribution, and is what we use to train the *black-box predictor* $b$. After initial training of $b$, we cannot modify or access its weights/architecture, re-train it, etc. We then *tune* the black-box predictions in a post-hoc manner in order to perform well on the out-of-distribution test set by using a (small) validation set $S$. Generally, the size of $|S| \ll$ the size of the training set, and so the practitioner stands to gain from using some of the power of $b$. Finally, we report results of the adapted model on the hold-out test set.

To simulate this setup in our experiments, in each setting we fix a certain model to be the "base black-box classifier" $b$. Then we investigate how much we can improve upon $b$ by only modifying its *predictions* and not the model itself.

To measure statistical significance and better understand how sensitive each evaluated method is to the individual samples which appear in the validation set $S$, we run each experiment multiple

---

[3]Assume for simplicity that $1/\epsilon$ is an integer.

times across a variety of validation set sizes. For any fixed sample size $n$, we sample five different validation sets $S$, and report the mean and standard deviation of each post-processing method across these five runs. The hold-out test set and base black-box predictor are always kept as fixed throughout. In particular, we only train the black-box predictor once — usually using the entire original training set — for all experiments.

To fairly compare our proposed CWPLUGIN method, we mostly consider baselines which are focused on post-hoc classifier adaptation, and do not require re-training the underlying model via importance weighing, invariant risk minimization, etc. (Arjovsky et al., 2019; Azizzadenesheli, 2021; Lipton et al., 2018). Nonetheless, we do consider training or fine-tuning a clean model from scratch on the validation set $S$ wherever applicable.

To the best of our knowledge, the only comparable *family* of post-hoc model adaptation techniques are calibration methods. This is because many calibration techniques are post-hoc and operate using *only* (multiclass) black-box predictions and true labels. Note, however, that most calibration techniques have goals slightly orthogonal to ours: they seek to increase accuracy or produce calibrated probabilities by minimizing the negative log likelihood (NLL) or similar quantities, and do not explicitly optimize for a particular metric of interest. On the other hand, CWPLUGIN takes as input the metric of interest (e.g., Accuracy, F-measure, etc.) and optimizes for it explicitly.[4]

We list out the baselines we include, deferring their additional implementation details to Appendix B.1.

**Clean.** Throughout, *clean* represents the raw hold-out test performance of the black-box predictor with no post-processing applied. If a method improves upon clean, then it means that the small validation set $S$ was helpful in adapting or improving the base black-box classifier $b$.

**Vector.** We include a variant of *vector scaling* (Guo et al., 2017), a standard in post-hoc calibration.

**Dirichlet Calibration.** Introduced by Kull et al. (2019), Dirichlet calibration is a family of methods which can be implemented directly on top of class probabilities. We include two versions amongst our baselines: **DiagDirich** and **FullDirich**, which roughly correspond to learning post-hoc estimators with $m$ weights for the former, and $m^2$ for the latter.

**Probing Classifier.** The post-hoc "probing classifier" approach from Hiranandani et al. (2021) can also take in an arbitrary (confusion matrix-based) metric as input and optimize for it. We use the authors' original implementation, but restrict to the version which does not use feature-defined groups in order to refine the estimates.

**Metrics Evaluated.** Note that only our CWPLUGIN method and the probing classifier method take as input the metric to be optimized as input. We generally run our experiments with Accuracy and *macro* variants of F-measure, G-mean, and Matthews Correlation Coefficient (MCC). We use the scikit-learn (Kramer & Kramer, 2016) F-measure implementation, the imbalanced-learn (Lemaître et al., 2017) implementation of G-mean, and our own implementation of MCC.

## 4.1 INCOME PREDICTION UNDER DISTRIBUTION SHIFT

We begin by experimenting with the ACSIncome dataset as made available by Ding et al. (2021), comprised of data from the US Census bureau in 2018. The predictive task we choose uses the provided features (Age, marriage status, education, etc.) in order to predict the income of each individual bucketed into one of $m = 3$ classes (income range in 0-30K, 30K-50K, or 50K+). The census data is also separated by state. We model distribution shift by training the black-box predictor as a simple linear regression (LR) model on 30K randomly drawn examples from California, and having our test set be 27K randomly drawn samples from Texas. We then vary the size of the validation set $S$ by randomly sampling an increasing number of datapoints from Texas. A subset of the results are in Figure 2; we defer the full set to Appendix B.2. We also include an additional baseline, **Logistic**, where we use the entire available validation set $S$ from Texas to fit a new logistic regression model on the target distribution.

Overall, our results here demonstrate that when the base (black-box) classifier has sufficient performance, a simple adaptation method such as CWPLUGIN or probing can provide a sizable performance boost with only a very small amount of tuning (validation) data. Importantly, both of these methods

---

[4]Notice that this implies the probabilities output by CWPLUGIN will in general *not* be calibrated.

even outperform training a logistic regression model from scratch on the validation set $S$. This indicates that there is some level of transferability between the two income prediction tasks between California and Texas, as expected.

| Method | F-measure | Accuracy |
|--------|-----------|----------|
| Clean | $0.483 \pm 0.000$ | $0.614 \pm 0.000$ |
| Logistic | $0.515 \pm 0.021$ | $0.610 \pm 0.005$ |
| Probing | $0.576 \pm 0.003$ | $0.614 \pm 0.000$ |
| Vector | $0.516 \pm 0.023$ | $0.617 \pm 0.002$ |
| FullDirich | $0.518 \pm 0.025$ | $0.616 \pm 0.002$ |
| DiagDirich | $0.516 \pm 0.023$ | $0.617 \pm 0.002$ |
| CWPLUGIN | $\mathbf{0.579 \pm 0.006}$ | $\mathbf{0.619 \pm 0.001}$ |

Figure 2: Distribution shift on US Census data; Mean and standard deviation across five validation set samples. (Left) Table showing test performance metrics at a validation set size of 50 samples. Using the proposed plugin method to adapt a classifier trained on California data to Texas data outperforms training a new classifier with only the (limited) available Texas data. (Right) Test F-measure performance across varying validation set size.

## 4.2 ADAPTING FINE-TUNED LANGUAGE MODELS

In this section, we evaluate how CWPLUGIN can help adapt and improve open-source language models in a variety of different language classification tasks. Throughout these tasks, we also include an additional baseline **BERT-FT**. This baseline represents an open-source pre-trained BERT model (Devlin et al., 2018) which is finetuned on the variable sized validation set $S$; we defer implementation details to Appendix B.1. This is certainly a reasonable solution that practitioner may prefer over using a closed-source black-box model. Indeed, with enough samples, we expect BERT-FT to outperform any purely black-box model post-processing domain adaptation technique such as CWPLUGIN. However, in the small sample regime ($|S| \leq 200\text{-}400$ samples), we demonstrate that simple and computationally cheap post-processing techniques learned on top of a black-box model can sometimes perform better.

### 4.2.1 SENTIMENT CLASSIFICATION

The first task we consider is **lmtweets**. As our baseline black-box predictor, we utilize a distilBERT-based model which was already fine-tuned on a variety of multilingual sentiment datasets, and uploaded to HuggingFace (Yuan, 2023). We evaluate the effectiveness of various post-processing methods on the tweet sentiment classification task introduced in SemEval-2017 (Rosenthal et al., 2017); this task is *out-of-distribution* for the trained model. The tweet sentiment classification task requires the model to predict the sentiment of a piece of language as one of three classes in the set {*positive, neutral, negative*}. A selection of results appear in Figure 3; we defer the full results to Appendix B.3. Note that BERT-FT represents a pre-trained BERT model which is only fine-tuned on the validation set $S$; this is separate from the distilBERT model trained on multilingual sentiments and used as our base black-box predictor.

In the **lmtweets** setting, we find that BERT-FT eventually outperforms all post-hoc adaptation methods at around $|S| = 400$. Nonetheless, CWPLUGIN is the best performing method best at sample sizes smaller than this. We also remark that it seems difficult for any post-processing method to improve upon base G-mean or Recall of the clean distilBERT (black-box) model; all post-processing methods fail to improve upon these base metrics on the hold-out test set. However, CWPLUGIN is the only method which *does not significantly harm* performance on these metrics.

### 4.2.2 EMOTION CLASSIFICATION

The second setting includes two tasks: **lmemotions** and **lmemotionsOOD**. As our black-box predictor for **lmemotions**, we utilize an open source DistilRoBERTa model which was trained on a variety of sentiment analysis tasks (Hartmann, 2022). The base model was trained to predict one of seven classes emotions {*anger, disgust, fear, joy, neutral, sadness, surprise*}. For **lmemotionsOOD**, we utilize a RoBERTa model trained on a variety of datasets of tweets as our black-box predictor (Camacho-Collados et al., 2022). The test set we evaluate both models performance on is the emotion

| Method | Accuracy | F-measure | G-mean | MCC |
|--------|----------|-----------|--------|-----|
| Clean | $0.452 \pm 0.000$ | $0.367 \pm 0.000$ | $\mathbf{0.621 \pm 0.000}$ | $0.232 \pm 0.000$ |
| Probing | $0.452 \pm 0.000$ | $0.328 \pm 0.023$ | $0.590 \pm 0.014$ | $0.148 \pm 0.011$ |
| Vector | $0.554 \pm 0.014$ | $0.448 \pm 0.037$ | $0.579 \pm 0.017$ | $0.227 \pm 0.027$ |
| FullDirich | $0.562 \pm 0.004$ | $0.470 \pm 0.037$ | $0.594 \pm 0.017$ | $0.255 \pm 0.007$ |
| DiagDirich | $0.554 \pm 0.014$ | $0.448 \pm 0.037$ | $0.579 \pm 0.017$ | $0.227 \pm 0.027$ |
| BERT-FT | $0.545 \pm 0.033$ | $0.391 \pm 0.033$ | $0.548 \pm 0.025$ | $0.191 \pm 0.059$ |
| CWPLUGIN | $\mathbf{0.563 \pm 0.003}$ | $\mathbf{0.504 \pm 0.003}$ | $0.619 \pm 0.013$ | $\mathbf{0.256 \pm 0.007}$ |

Figure 3: Mean and standard deviation across five validation set samples. (Top) **lmtweets** results for each method on each metric using a sized 160 validation set $S$. (Bottom) **lmtweets** test G-mean and F-measure performance across varying validation set size. Adapting the outputs of a black-box model with CWPLUGIN outperforms other post-hoc adaptation techniques at $\leq 400$ samples. At $\geq 400$ samples, fine-tuning a clean BERT model on the validation set (BERT-FT) starts performing better.

classification dataset introduced by Saravia et al. (2018). This task asks the model to predict one of six of the seven emotions listed previously.

Importantly, the emotion classification dataset *was included in the original fine-tuning data* of the model for **lmemotions**. A performance improvement here would indicate that any post-processing methods can help *specialize* a model on a subset of its own training data on specific metrics of interest. On the other hand, the emotion classification dataset was *not* included for **lmemotionsOOD**; hence, the task is *out-of-distribution*.

A selection of results for both settings appear in Figure 4; full results are in Appendix B.4. At a high level, CWPLUGIN performs favorably relative to the calibration and probing approaches on the tested metrics in both settings. **lmemotions** is of particular interest; since the validation and test data are *in-distribution*, but our results showcase the fact that the optimal predicted probabilities may be significantly altered when considering metric optimization rather than accuracy (or calibration) error minimization. Since each method tested relies only on the predictions of the model, a practitioner may see benefit from a "plug-and-play" approach in which different post-hoc estimators are learned and applied to different settings with different metric optimization requirements.

### 4.3 ADAPTING LANGUAGE MODELS IN NOISY DOMAINS

In this section, we show that CWPLUGIN can also perform well in the presence of label shift (Lipton et al., 2018; Storkey, 2008) or label noise (Natarajan et al., 2013; Patrini et al., 2017). Let $D'$ be the source distribution, and $D$ the target distribution. We test for learning under *knock-out* label shift. This setting is motivated by, for example, disease classification, where during an outbreak $D(y|x)$ may be larger than historical data $D'(y|x)$, but the manifestations of the disease $D(x|y) = D'(x|y)$ may not change (Lipton et al., 2018). In our experiments, we model label shift by randomly deleting a fraction of a subset of classes in $D$ relative to the original source distribution $D'$. We also test for symmetric, class-dependent label noise. That is, for a certain subset of classes, datapoints of that class have their labels in the validation set $S$ flipped to another class — chosen uniformly at random — with probability $p$.

We test these two types of noise on two language classification tasks: SNLI (Bowman et al., 2015) and ANLI (Nie et al., 2020). For both the label shift and label noise settings, we utilize a model trained on GLUE (Wang et al., 2019) and ANLI as our base, black-box predictor (Wong, 2023; Li et al., 2023). Details about the specific parameters of label noise and label shift are deferred to

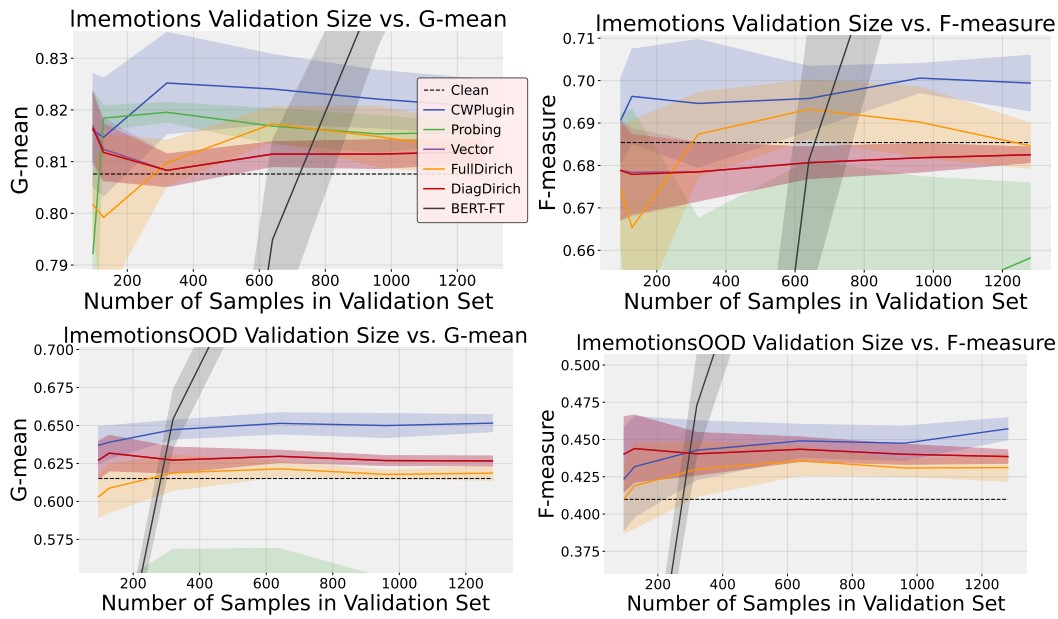

Figure 4: Mean and standard deviation across five runs. Results for **Imemotions** (top) and **ImemotionsOOD** (bottom) on G-mean and F-measure. CWPLUGIN consistently performs well across metrics for smaller sample sizes relative to all tested baseline methods including fine-tuning a clean language model on only the validation set (BERT-FT).

| Method | F-measure | G-mean | | Method | F-measure | G-mean |
|---|---|---|---|---|---|---|
| Clean | $0.575 \pm 0.000$ | $0.656 \pm 0.000$ | | Clean | $0.276 \pm 0.000$ | $0.528 \pm 0.000$ |
| Probing | $0.589 \pm 0.025$ | $0.723 \pm 0.008$ | | Probing | $0.264 \pm 0.033$ | $0.505 \pm 0.037$ |
| Vector | $0.590 \pm 0.020$ | $0.681 \pm 0.018$ | | Vector | $0.331 \pm 0.060$ | $0.516 \pm 0.028$ |
| FullDirich | $0.578 \pm 0.037$ | $0.678 \pm 0.013$ | | FullDirich | $0.365 \pm 0.027$ | $0.524 \pm 0.017$ |
| DiagDirich | $0.590 \pm 0.020$ | $0.681 \pm 0.018$ | | DiagDirich | $0.331 \pm 0.060$ | $0.516 \pm 0.028$ |
| CWPLUGIN | $\mathbf{0.613 \pm 0.011}$ | $\mathbf{0.724 \pm 0.018}$ | | CWPLUGIN | $\mathbf{0.406 \pm 0.008}$ | $\mathbf{0.541 \pm 0.015}$ |

Figure 5: (Left) Results for SNLI with label shift applied to the validation and test data for methods fit on $|S| = 100$ validation samples. (Right) Results for ANLI with label noise on $|S| = 250$ validation samples. In both cases, CWPLUGIN performs favorably when compared to other baselines.

Appendix B.5; a summary of the results is given in Figure 5. Overall, these experiments demonstrate that our proposed CWPLUGIN method can also be useful in adapting black-box models to varying degrees of test-time or train-time noise.

## 5 LIMITATIONS AND CONCLUSIONS

One limitation of CWPLUGIN is that it may be very dependent on the available number of samples for the selected *fixed* class. Throughout our discussion, we chose class $m$ as the fixed class; however, in practice we found that choice of this fixed class can impact performance and the ability to fit a meaningful signal in the data. Another limitation is that since the post-processing method utilizes solely the probabilistic multiclass predictions — and not any feature information — the *quality* of these predictions is quite important in determining the outcome of the method. For example, predictions which are not calibrated, or do not represent meaningful probabilities may allow for less expressiveness of post-hoc estimators, which limits this class of post-processing methods. We leave investigating both these directions more rigorously to future work.

We believe that our work represents an important direction in the ever-changing model marketplace. As black-box predictors potentially become more common solutions to machine learning practitioner application domains, post-hoc methods like CWPLUGIN may eventually allow practitioners some degree of model adaptation to particular tasks of interest.

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

# A PROOFS

## A.1 RESULTS FROM MAIN TEXT

*Proof of Lemma 3.* It is a standard fact that quasi-concavity of $f$ over a certain restricted domain — here, $\alpha \in [0, 1 - \epsilon]$ — implies that $f$ is *uni-modal* over said domain (see, e.g., Boyd & Vandenberghe (2004, Ch. 3.4)). Requiring $f$ to be quasi-concave when restricted to any pair of classes $k, m$ — formally, $f(\mathbf{C}^{h_\alpha^{k,m}})$ is quasi-concave for all $k, m$ — therefore implies that binary search will be optimal up to an additive $\epsilon/2$ factor. $\square$

## A.2 FULL ANALYSIS OF CWPLUGIN METHOD

In this section, we show that in the weight elicitation framework, CWPLUGIN can be used to find the optimal weights for linear-diagonal metrics. We begin our theoretical results by defining *linear-diagonal* metrics.

**Definition 4** (Linear Diagonal Metric). *A metric of the confusion matrix* $f : \mathbf{C}^h \mapsto \mathbb{R}_{\geq 0}$ *is* linear diagonal *if it can be written as* $f(\mathbf{C}^h) = \sum_{i=1}^m \beta_i \cdot \mathbf{C}_{i,i}^h$ *for* $\|\beta\|_1 = 1$.

This captures, for example, accuracy and weighted accuracy.

We first prove that CWPLUGIN is a *consistent* classifier, in that it will recover the Bayes optimal predictor for any linear diagonal metric, when working with the relevant population-level quantities. First, we make the following assumption on the conditional label distribution $\eta(x)$.

**Assumption 5.** *Let a ground truth distribution $D$ supported on $\mathcal{X} \times \mathcal{Y}$ be given. Assume that the true conditional label distribution $\eta(x)$ satisfies that for any pair of classes $k, k'$, we have that the function* $\mathbb{P}_{x \sim D_{\mathcal{X}}} \left[ \frac{\eta(x)_k}{\eta(x)_{k'}} \geq t \right]$ *is continuous and strictly decreasing for all* $t \in [0, \infty)$.

This is a multiclass generalization of a standard measurability assumption from binary classification that thresholding events have positive density but non-zero probability (see, e.g., Assumption 1 of Hiranandani et al. (2019a)). This assumption is satisfied by many smooth predictors, including, for example, any softmax predictor.

We are now ready to state our consistency result. Intuitively, this result states that the CWPLUGIN method learns the correct weights $\mathbf{w} = \beta$ when run on the population quantities (infinite samples and with access to the true class-conditional probability distribution $\eta$), and with only *query* access to the metric $f$. Furthermore, the resulting classifier using the weighted predictions (the final line **Inference** of Algorithm 1) is indeed Bayes optimal.

**Proposition 6.** *Let a ground truth distribution $D$ supported on $\mathcal{X} \times \mathcal{Y}$ be given, and assume that the conditional label distribution $\eta(x)$ satisfies Assumption 5. Let coefficients $\beta$ define a linear diagonal performance metric* $f(\mathbf{C}^\eta) = \sum_{k=1}^m \beta_k \mathbf{C}_{k,k}^\eta$, *and ensure that* $\|\beta\|_1 = 1$. *Suppose that we vary the searched weights* $\alpha \in [0, 1 - \rho]$ *in line 5 of Algorithm 1 through* $\rho = \min_{k \in [m-1]} \frac{\beta_m}{\beta_m + \beta_k} > 0$. *Then, the weights $\mathbf{w}$ learned (elicited) by running CWPLUGIN with the population quantities will be equivalent to the weights for the Bayes optimal predictor for the metric $f$.*

*Proof.* Without loss of generality, assume that $\beta_m > 0$; if it wasn't, choose any other index $j \neq m$ s.t. $\beta_j > 0$. If there is no such index, the metric is trivial (all zero weights, contradiction). Let $\beta \in \rho(m)$ correspond to the true metric weights. Then, consider the (normalized) weights $\overline{\beta_k} = \beta_k/\beta_m$, which gives the optimal relative weight between class $k$ and $m$.

Notice that $\overline{\beta} = (\overline{\beta_1}, \overline{\beta_2}, \ldots, \overline{\beta_{m-1}}, 1)$. Furthermore, for $0 < \rho = \min_{k \in [m-1]} \frac{\beta_m}{\beta_m + \beta_k}$ small enough, for all $k \in [m - 1]$ there exists $\alpha_k^* \in [0, 1 - \rho]$ such that $\overline{\beta_k} = \beta_k/\beta_m = \frac{\alpha_k^*}{1 - \alpha_k^*}$. In particular, solving the equation gives us that $\alpha_k^* = \frac{\beta_k}{\beta_m + \beta_k} \in [0, 1 - \rho]$. This allows us to write out the following equivalent form of $\overline{\beta}$:

$$\overline{\beta} = (\overline{\beta_1}, \overline{\beta_2}, \ldots, \overline{\beta_{m-1}}, 1) = \left( \frac{\beta_1}{\beta_m}, \ldots, \frac{\beta_{m-1}}{\beta_m}, 1 \right) = \left( \frac{\alpha_1^*}{1 - \alpha_1^*}, \ldots, \frac{\alpha_{m-1}^*}{1 - \alpha_{m-1}^*}, 1 \right)$$

We want to show that the $\mathbf{w}$ output by CWPLUGIN (pre-normalization) is exactly $\overline{\beta}$.

Fix a class pair $(k, m)$, and recall the definition of the restricted classifier for that pair:

$$h_\alpha(x) = h_\alpha^{k,m}(\eta(x)) = \begin{cases} k & \text{if} \quad \alpha\eta(x)_k > (1-\alpha)\eta(x)_m \\ m & \text{otherwise.} \end{cases} \tag{2}$$

If we can prove that $h_\alpha(x)$ is identical to the Bayes optimal classifier for $f$ *restricted* to

Next, consider the metric evaluated at the *population* confusion matrix for $h_\alpha$.

$$\begin{aligned} f(\mathbf{C}^{h_\alpha}) &= \beta_k \mathbf{C}_{k,k}^{h_\alpha} + \beta_m \mathbf{C}_{m,m}^{h_\alpha} \\ &= \frac{\beta_k}{\beta_m} \mathbf{C}_{k,k}^{h_\alpha} + \mathbf{C}_{m,m}^{h_\alpha} \\ &= \frac{\alpha_k^*}{1-\alpha_k^*} \mathbf{C}_{k,k}^{h_\alpha} + \mathbf{C}_{m,m}^{h_\alpha} \\ &= \frac{\alpha_k^*}{1-\alpha_k^*} \mathbb{E}_{(x,y)\sim D}\left[\mathbf{1}[h_\alpha(x) = k] \cdot \mathbf{1}[y = k]\right] + \mathbb{E}_{(x,y)\sim D}\left[\mathbf{1}[h_\alpha(x) = m] \cdot \mathbf{1}[y = m]\right] \\ &= \mathbb{E}_{(x,y)\sim D}\left[\frac{\alpha_k^*}{1-\alpha_k^*}\mathbf{1}[h_\alpha(x) = k] \cdot \mathbf{1}[y = k] + \mathbf{1}[h_\alpha(x) = m] \cdot \mathbf{1}[y = m]\right] \end{aligned} \tag{3}$$

We claim that the $h_\alpha$ which maximizes this quantity is precisely the $h_\alpha$ defined by $\alpha = \frac{\beta_k}{\beta_m + \beta_k}$. To prove this, we appeal to the following Lemma. Note that this lemma is stated for the *binary case* $\mathcal{Y} = \{0, 1\}$, where $\eta^{\text{bin}}(x) \in [0, 1]$ instead of $\Delta(m)$.

**Lemma 7** (Proposition 2 Hiranandani et al. (2019a)). *Let a ground truth distribution $D$ over $\mathcal{X} \times \{0, 1\}$ be given. Assume that the conditional label distribution $\eta^{\text{bin}}(x)$ has the property that $\mathbb{P}_{x\sim \mathbf{D}_\mathcal{X}}[\eta^{\text{bin}}(x) \geq t]$ is continuous and strictly decreasing for $t \in [0, 1]$. Then, for any linear diagonal metric $f(\mathbf{C}^h) = \beta_1 \mathbf{C}_{1,1}^h + \beta_2 \mathbf{C}_{2,2}^h$, the RHS in Equation (3) is maximized by $\alpha = \beta_1/(\beta_1 + \beta_2)$.*

We can apply this result because of Assumption 5 being a strictly more general version of the assumption required by the lemma. The proof of this lemma is in fact a technical insight in the proof of part 2 of Proposition 2 in Hiranandani et al. (2019a). Ultimately, it is true because the boundary of the set of all confusion matrices can be characterized by a family of *threshold classifiers* (Lemma 2 in Hiranandani et al. (2019a)), of which the optimal value for $\alpha$ can be explicitly optimized over by taking a simple derivative. The boundary of the set of all confusion matrices is the only important quantity since we know it contains all classifiers which have optimal metric value (since $f$ is a linear function).

Using this lemma, we have that the optimal restricted classifier will be given by $h^\alpha$ defined with $\alpha = \frac{\beta_k}{\beta_m + \beta_k}$. Notice, however, that $\alpha_k^* = \alpha$. Therefore, applying the argument across all pairs of classes suffices to prove that we recover the underlying linear diagonal metric weights $\overline{\beta}$. Finally, re-normalizing (line 8 of Algorithm 1) then implies we have recovered the original weights $\beta$.

To show that the *predictor* recovered by weighing the predictions with $\mathbf{w} = \beta$ as done in the final line of Algorithm 1:

$$h_{\text{plugin}}^{\mathbf{w}}(x) = \arg\max_{k\in[m]} b(x)_k \mathbf{w}_k,$$

is indeed a Bayes-optimal predictor, we conclude with the following standard result.

**Lemma 8** (Prop. 5 of Narasimhan et al. (2023)). *Any predictor $h^*$ of the following form is a (consistent) Bayes optimal classifier for a linear diagonal metric $f$ with diagonal weights $\beta_i$: $h^*(x) \in \arg\max_{i\in[m]} \beta_i \cdot \eta(x)_i$.*

This concludes the proof. $\qquad\square$

We note that an equivalent result could have been proven by utilizing a certain restricted Bayes optimal classifier lemma from prior work (Hiranandani et al., 2019b, Proposition 2).

Next, we will utilize the consistency result in order to obtain a finite sample guarantee. That is, with only a finite number of samples of the true class-conditional label distribution, we can still (approximately and w.h.p.) obtain the underlying metric weights for $f$ given by $\beta$.

**Proposition 9.** *Let $f(\mathbf{C}^h) = \sum_{k=1}^m \beta_k \mathbf{C}_{k,k}^h$ be a linear diagonal metric with $\|\beta\|_1 = 1$. Fix a failure probability $\delta \in (0, 1)$. Suppose that $\alpha$ is obtained to precision $\epsilon$ in line 5 of Algorithm 1. This can be done via a line or binary search to precision $\epsilon$ over the boundary $\alpha \in [0, 1 - \rho]$ for $\rho = \min_{k \in [m-1]} \frac{\beta_m}{\beta_m + \beta_k} > 0$. Then, with probability at least $1 - \delta$ over sample $S = \{(\eta(x_i), y_i)\}_{i \in [n]}$ where $(x_i, y_i) \sim D$ i.i.d., the coefficients $\mathbf{w}$ output by Algorithm 1 satisfy:*

$$\|\beta - \mathbf{w}\|_1 \le O\left(m \cdot \frac{\gamma}{(1-\rho)^2}\right) \quad \text{for } \gamma = C\sqrt{\frac{\log(1/\delta)}{n}} + \epsilon/2,$$

*for some positive constant $C > 0$.*

*Proof.* Let $\beta$ denote the true weight coefficients of $f$ (unavailable to the learner). Let $\beta^S$ denote the optimum weights maximizing the metric $f$ on the sample $S$, and let $\mathbf{w}$ denote the weights output by CWPLUGIN in Algorithm 1. We will instead work with the un-normalized quantities $\overline{\beta}$, $\overline{\beta^S}$, and $\overline{\mathbf{w}}$, which have the property that $\overline{\beta_k} = \beta_k/\beta_m$, e.g.,

$$\overline{\beta} = (\overline{\beta_1}, \overline{\beta_2}, \ldots, \overline{\beta_{m-1}}, 1).$$

Similarly for $\overline{\beta^S}$ and $\overline{\mathbf{w}}$.

Without loss of generality, assume that $\overline{\beta_m} = \overline{\beta_m^S} = \overline{\mathbf{w}_m} = 1$. By construction (see proof of Proposition 6), for any class $k \ne m$ we know that there exists $\alpha_k^*, \alpha_k^S, \alpha_k \in [0, 1 - \rho)$ such that:

$$\overline{\beta_k} = \beta_k/\beta_m = \frac{\alpha_k^*}{1 - \alpha_k^*}$$

$$\overline{\beta_k^S} = \beta_k^S/\beta_m^S = \frac{\alpha_k^S}{1 - \alpha_k^S}$$

$$\overline{\mathbf{w}_k} = \mathbf{w}_k/\mathbf{w}_m = \frac{\alpha_k}{1 - \alpha_k}$$

We bound the relationship between $\alpha$s as follows.

$$|\alpha_k^* - \alpha_k| \le |\alpha_k^* - \alpha_k^S| + |\alpha_k^S - \alpha_k| \le C\sqrt{\frac{\log(1/\delta)}{n}} + \epsilon/2 = \gamma \tag{4}$$

For some constant $C > 0$. We bound the first term in the second inequality by Hoeffding's, and the second term by the Proposition 6 and the fact that due to the granularity of the line search in Algorithm 1, we know that $|\alpha_k - \alpha_k^S| \le \epsilon/2$.

Finally, we can bound the weight difference for any class $k$ as follows.

$$|\overline{\beta_k} - \overline{\mathbf{w}_k}| \le |\overline{\beta_k} - \overline{\beta_k^S}| + |\overline{\beta_k^S} - \overline{\mathbf{w}_k}| \le \left|\frac{\alpha_k^*}{1 - \alpha_k^*} - \frac{\alpha_k^S}{1 - \alpha_k^S}\right| + \left|\frac{\alpha_k^S}{1 - \alpha_k^S} - \frac{\alpha_k}{1 - \alpha_k}\right|$$

$$= \left|\frac{\alpha_k^* - \alpha_k^*\alpha_k^S - (\alpha_k^S - \alpha_k^*\alpha_k^S)}{(1 - \alpha_k^*)(1 - \alpha_k^S)}\right| + \left|\frac{\alpha_k^S(1 - \alpha_k) - \alpha_k(1 - \alpha_k^S)}{(1 - \alpha_k^S)(1 - \alpha_k)}\right|$$

$$\le \left|\frac{\alpha_k^* - \alpha_k^S}{(1 - \alpha_k^*)(1 - \alpha_k^S)}\right| + \left|\frac{\alpha_k^S - \alpha_k}{(1 - \alpha_k^S)(1 - \alpha_k)}\right|$$

$$\le 2 \cdot \frac{\gamma}{(1-\rho)^2}$$

In the last step, we used the fact from Equation (4) to bound the numerators by $\gamma$. For the denominators, note that each of $\alpha_k^*, \alpha_k^S, \alpha_k \in [0, 1 - \rho)$. Applying this to each $\overline{\beta_k}$ by triangle inequality completes the proof. □

# B  ADDITIONAL EXPERIMENT AND DATASET DETAILS

## B.1  ADDITIONAL BASELINE DETAILS

Here we give additional implementation details for the baseline methods we compare against. Post-hoc multiclass calibration techniques fall into two categories: techniques which operate on *logits* (raw, unscaled probabilities), and techniques which take as input class probabilities. We assume that only class probabilities are available to us as outputs of black box models, and as such, we mainly focus on the latter.

**Vector Scaling.** Let $\sigma_{\text{SM}} : \mathbb{R}^m \to \Delta_m$ be the softmax function. Given a black-box predictor $b$, *vector scaling* (Guo et al., 2017) learns a transformed estimator of $b$, given by $\sigma_{\text{SM}}(\boldsymbol{W} \cdot b(x_i) + \boldsymbol{c})$. The weight matrix $\boldsymbol{W} \in \mathbb{R}^{m \times m}$ and bias vector $\boldsymbol{c} \in \mathbb{R}^m$ are chosen in order to minimize the NLL on the calibration set. Note that the weight matrix $\boldsymbol{W}$ is restricted to be diagonal, and hence, the method is essentially learning $2m$ parameters. Furthermore, the original formulation actually fits the parameters on top of the model *logits*, which are unavailable to us. We modify the formulation to fit the class probabilities given as the output of $b(x_i)$. We use the vector scaling implementation given by NetCal in Küppers et al. (2020), which uses cross validation to select the best internal parameters.

**Dirichlet Calibration.** Introduced by Kull et al. (2019), Dirichlet calibration is a family of methods which can be implemented directly on top of class probabilities. The method is built on the assumption that the underlying prediction vectors are sampled from a Dirichlet distribution. Formally, Dirichlet calibration also learns a weight matrix $\boldsymbol{W}$ and bias $\boldsymbol{c}$ learn a classifier given by $\sigma_{\text{SM}}(\boldsymbol{W} \cdot \ln b(x_i) + \boldsymbol{c})$. In order to choose appropriate $\boldsymbol{W}$ and $\boldsymbol{c}$, Dirichlet calibration minimizes log loss combined with Off-Diagonal and Intercept Regularisation (ODIR). ODIR takes two hyperparameter values: $\lambda$ and $\mu$. We search over all combinations of $(\lambda, \mu) \in \{10^{-1}, 10^{-2}, 10^{-3}, 10^{-4}, 10^{-5}\}^2$. We select the best performing hyperparameter pair on the validation set $S$.

Throughout our experiments, we noticed similar performance of Diagonal Dirichlet and Vector scaling, even though the implementations are very separate. Given that we select the optimal Diagonal Dirichlet calibrator based on performance on the validation set $S$, the resulting solution may look nearly identical to Vector scaling at *smaller* regularization values. As the larger regularization values were rarely selected, the performance and optimized solution of both methods are quite similar.

**Probing Classifier.** In addition to calibration measures, we also report the performance of the "probing classifier" introduced in Hiranandani et al. (2021). This classifier is constructed via post-processing a black-box predictor $b$ by learning $m$ class weights, similar to plugin. However, these $m$ weights are found by solving a particular linear system which maximizes the metric of interest. We use the authors' original implementation, but restrict to the version which does not use feature-defined groups in order to refine the estimates. The method also takes in a *step-size* parameter $\epsilon$. We select the best performing parameter amongst $\epsilon \in \{0.1, 0.05, 0.01, 0.005, 0.001\}$ by taking the one with the best metric value on the (validation) set $S$.

**BERT-FT.** Our fine-tuning baseline takes the original open-source BERT-cased model from HuggingFace (Devlin et al., 2018), and fine-tunes it using AdamW on the validation set $S$ with batch size 64 over 100 epochs. We use a linear learning rate decay which kicks in after 500 warmup steps, and also utilize the default pre-trained BERT-cased tokenizer. We select the best performing model across all epochs (using only the set $S$, not any hold-out data). Then, we report the predictions of the model on the hold-out test set.

## B.2  INCOME PREDICTION EXPERIMENTS

We show the performance of all methods for Accuracy, F-measure, G-mean, MCC (Matthews Correlation Coefficient), and Fowlkes-Mallows Score (Fowlkes & Mallows, 1983) when scaling the number of samples in the validation set $S$. The results are in Figure 6.

Overall, we find that CWPLUGIN and probing are generally the best performing methods across different metrics, significantly outperforming the other methods on F-measure, G-mean, and Fowlkes-Mallows Score. We do not observe much improvement over the "clean" baseline for accuracy or MCC.

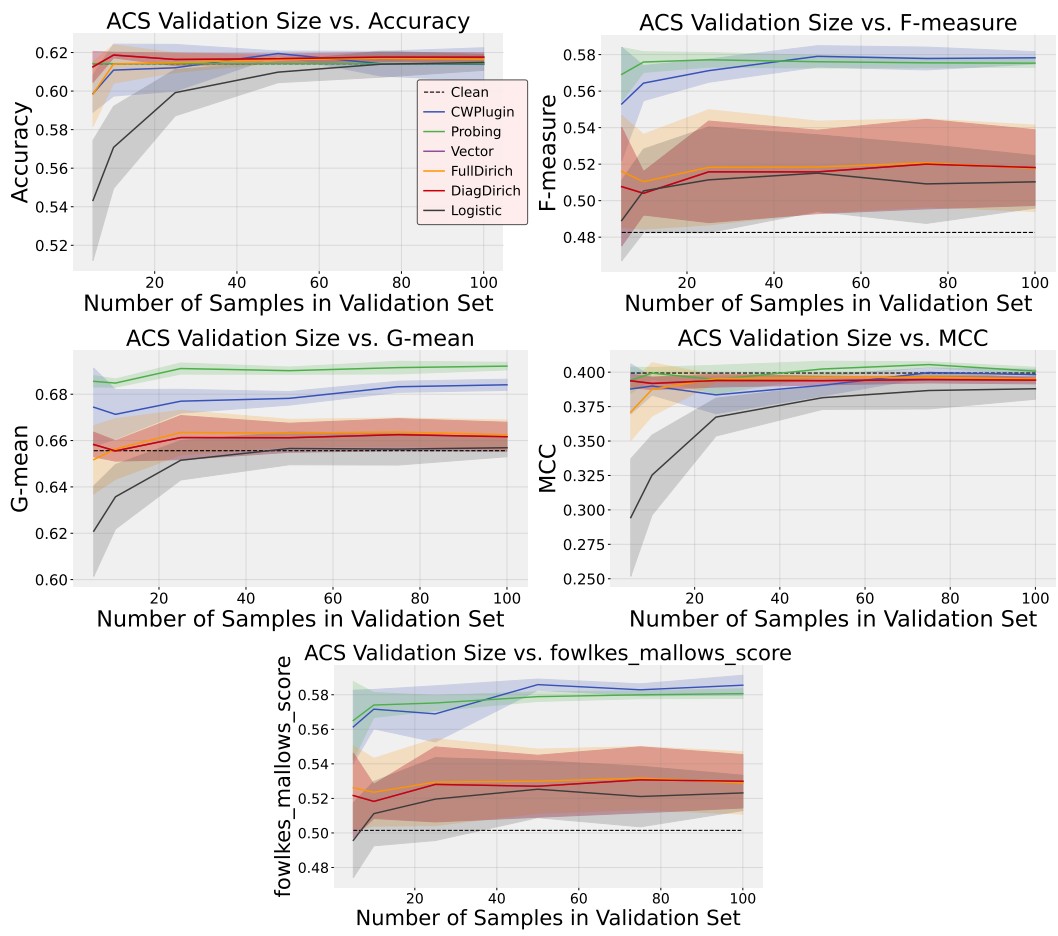

Figure 6: Performance of each method on each metric of ACSIncome.

### B.3 TWEET CLASSIFICATION EXPERIMENTS

To evaluate the black-box classifier described in the main text, we test its performance on the tweet sentiment classification dataset (Rosenthal et al., 2017). We use the hugging face datasets library to load the dataset using the function "cardiffnlp/super_tweeteval" for the task "tweet_sentiment". We use the entire "train" split of 16K examples, randomly splitting 20% into a hold-out test set. We then vary the size of the validation set through the remaining 80% of the examples.

The full performance of each method on each metric for the **lmtweets** task is shown in Figure 7.

### B.4 EMOTION CLASSIFICATION EXPERIMENTS

We use hugging face to access the "'dair-ai/emotion" dataset. We use the "train" split, which has 12.8K examples. We reserve 20% as our hold-out test set, and vary the validation set amongst the remaining 80%.

The full performance of each method on the **lmemotions** and **lmemotionsOOD** task is shown in Figure 8 and Figure 9.

### B.5 LANGUAGE SENTIMENT CLASSIFICATION EXPERIMENTS WITH NOISY LABELS

We utilize the SNLI and ANLI datasets as made available by HuggingFace datasets; these datasets each have three classes (positive, neutral, negative) which we refer to as class 0, 1, and 2.

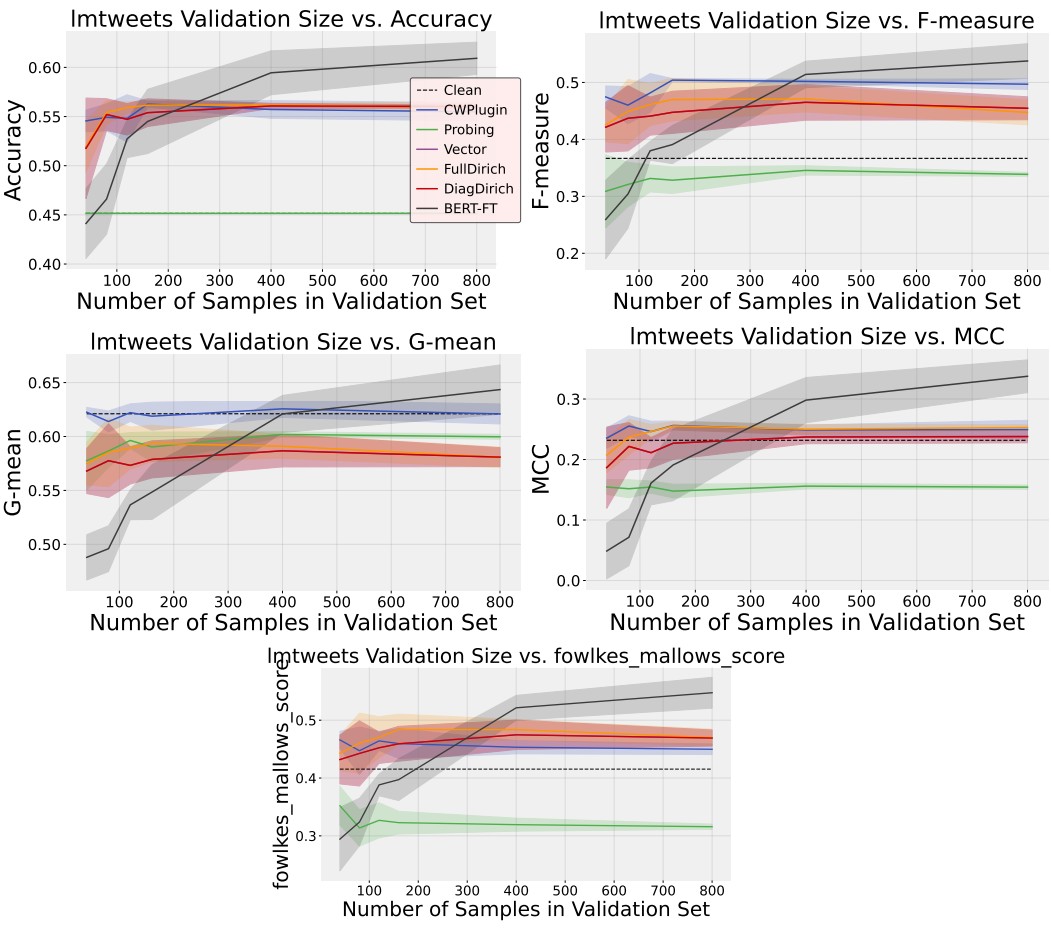

Figure 7: Performance of each method on each metric of **lmtweets**.

There are two settings, label *shift* and label *noise*. For label shift, we (randomly) delete 80% of both validation and test data with labels 0 or 1. For label noise, we flip each data point with true class 0 to a randomly chosen other class with 60% probability.

In each of the following cases, we save 20% for a holdout test set, and vary the validation set amongst the remaining 80%.

For ANLI labelshift, we utilize the "train" split, which has 10K examples. The full results are available in Figure 10. For SNLI labelshift, we utilize the "test" split, which has 10K examples. The full results are available in Figure 11. For ANLI labelnoise, we utilize the "test" split. The full results are in Figure 13. Finally, for SNLI label noise, we utilize the "train" split; the results are in Figure 12.

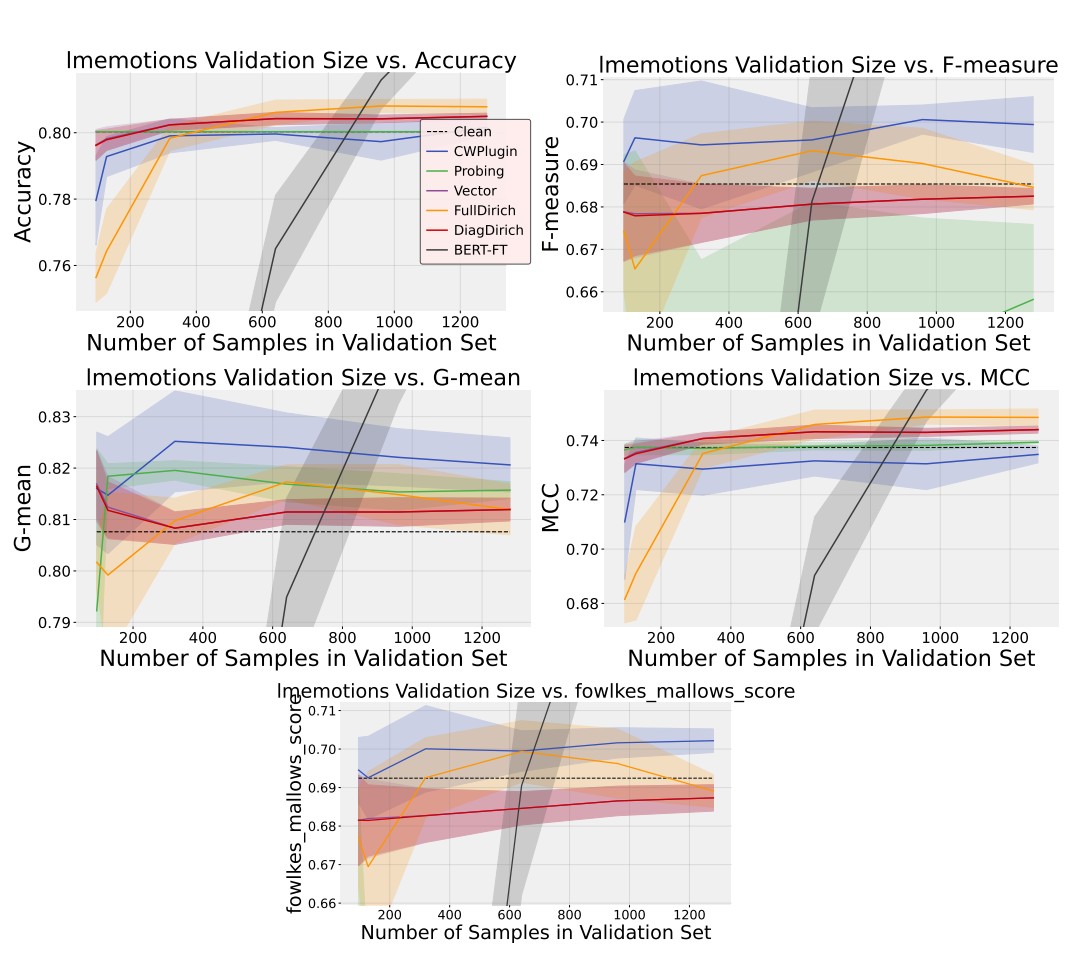

Figure 8: Performance of each method on each metric of **lmemotions**.

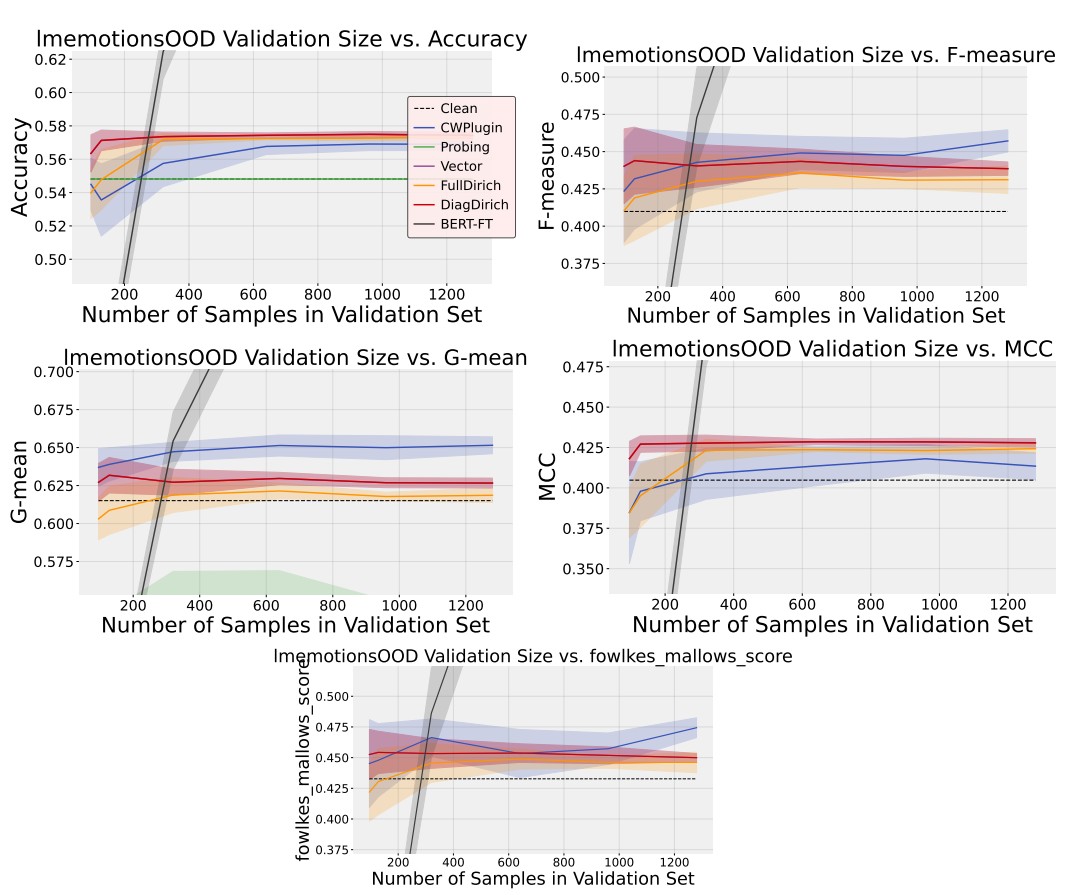

Figure 9: Performance of each method on each metric of **lmemotionsOOD**.

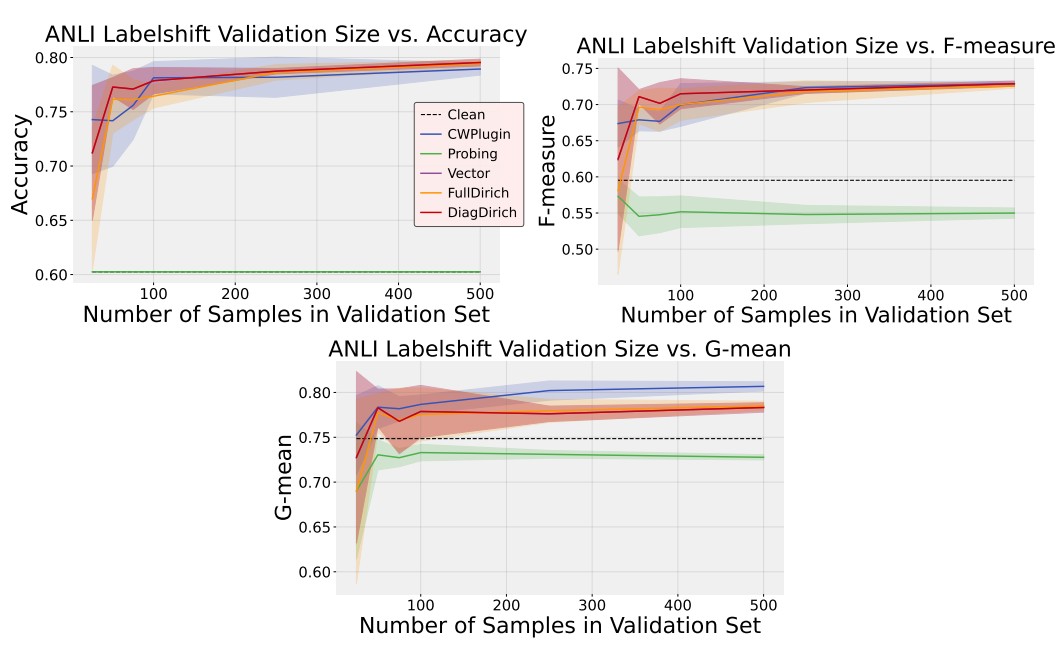

Figure 10: Performance of each method on each metric of ANLI with label shift.

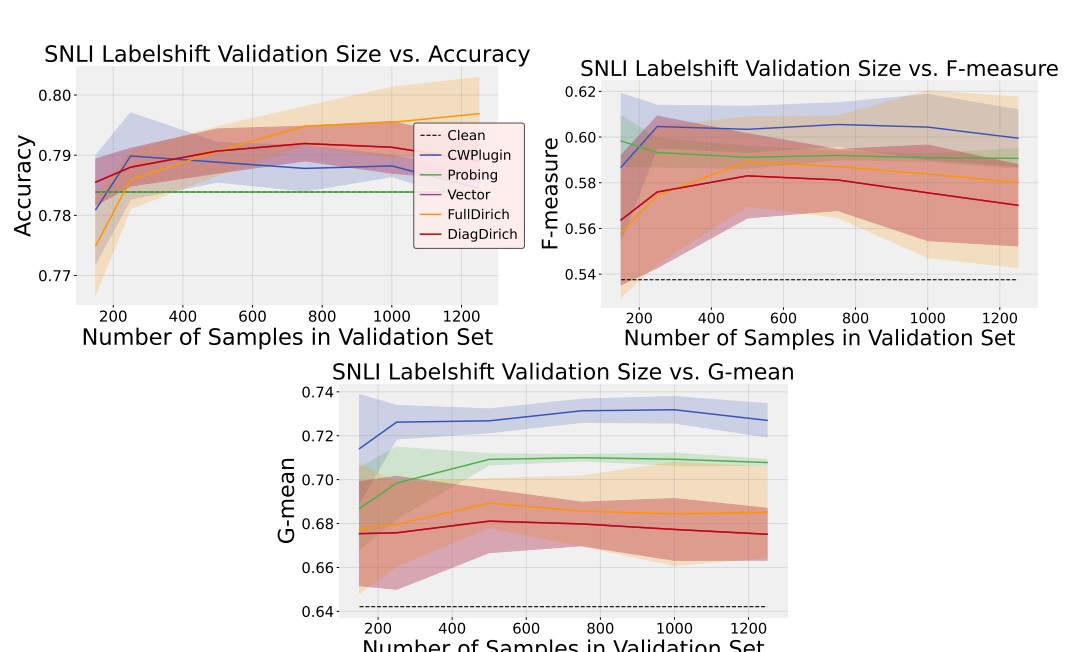

Figure 11: Performance of each method on each metric of SNLI with label shift.

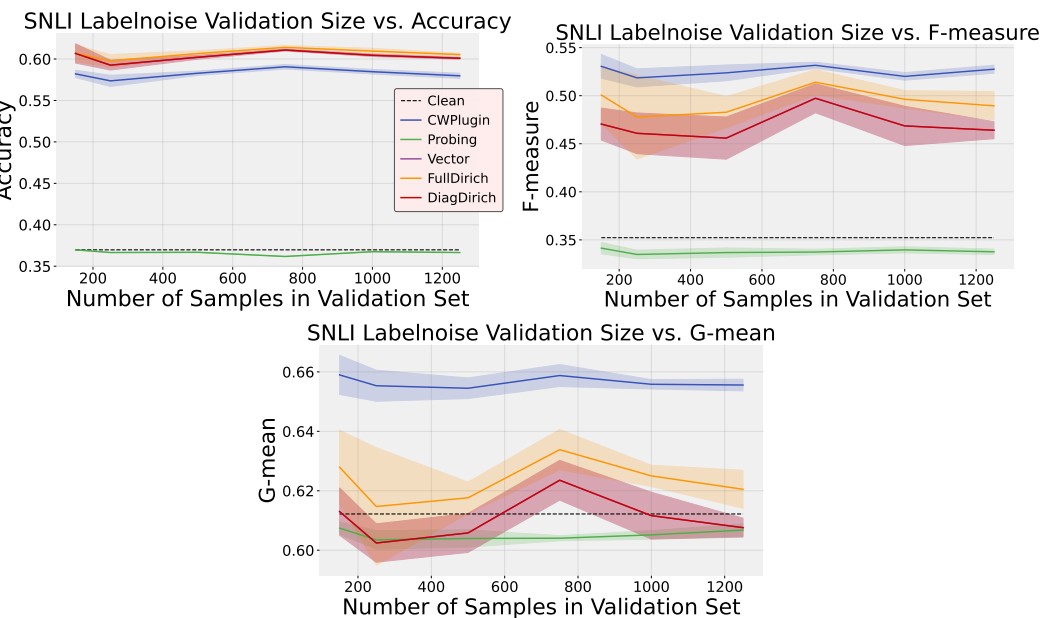

Figure 12: Performance of each method on each metric of SNLI with label noise.

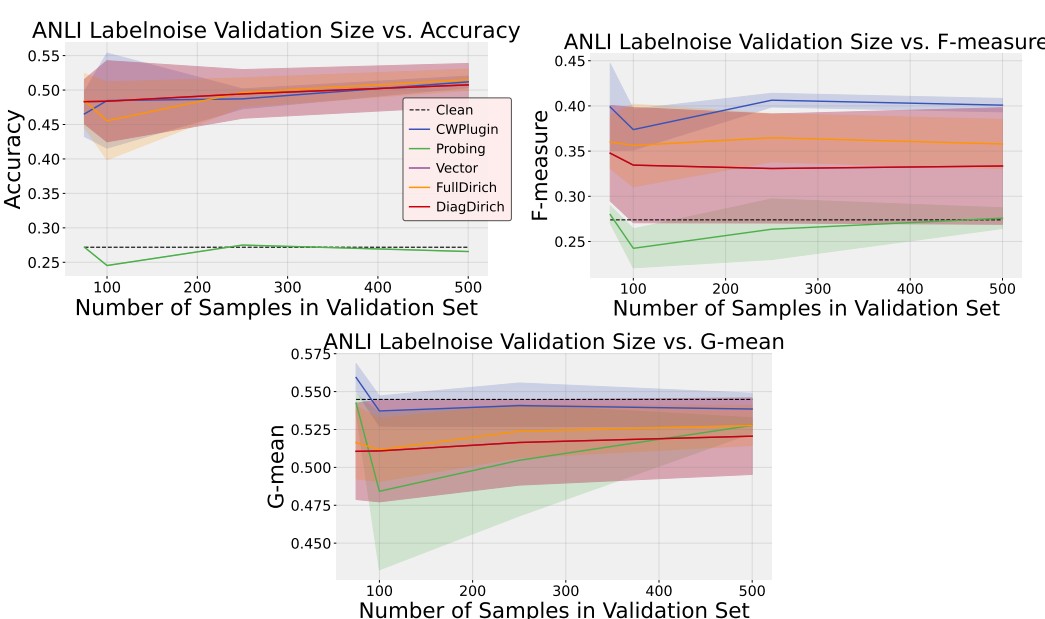

Figure 13: Performance of each method on each metric of ANLI with label noise.

