# OpenReview forum: "An Efficient Plugin Method for Metric Optimization of Black-Box Models"
_ICLR.cc/2025/Conference — ICLR 2025 Conference Withdrawn Submission_

### Official Review · Reviewer_KoWZ · 2024-10-30

**Soundness:** 3
**Presentation:** 4
**Contribution:** 2
**Rating:** 3
**Confidence:** 4

**Summary:**

This paper proposes a method to perform post-hoc adaptation of a black-box classifier, by tuning class weights to optimize a metric of interest on a target distribution. Specifically, the authors introduce CWPlugin, which performs a line search to find the best reweighting coefficient for each (m-1) class, based on pairwise comparisons against a reference class.

**Strengths:**

1. This paper focuses on an interesting and relevant problem, namely post-hoc adaptation of a black-box classifier, with only a small sample from a target distribution of interest.
2. The method proposes a novel way to optimize weights to re-weigh predicted probabilities and applies to multi-class classification.
3. The paper reads well, with intuition and formal depictions interleaved in a nice way.

**Weaknesses:**

**Main concerns:**
1. The authors only compare empirically against calibration techniques. The proposed problem of finding a set of weights $\mathbf{w}$ to adapt a pre-trained model to optimize a black-box metric (which the authors refer to as `query-access only` metrics) is an **instantiation of black-box optimization**. As such, mature BBO techniques, including Bayesian optimization [R1], population-based search (e.g., evolution strategies) [R2], and even random search, should be included in the empirical comparisons.
2. The proposed line-search method has quite high complexity $\mathcal{O}(mn/\varepsilon)$, and scales linearly with number of classes $m$, and number of samples $n$, and inversely with search resolution $\varepsilon$. Techniques like BO and ES remove this dependence on $m$, $n$ and might be more suitable. They also do not require the pairwise comparison that is the basis of the line search.
3. Fundamentally and theoretically, what are the benefits of the author's proposed approach over these BBO techniques (including BO, ES)? Based on my understanding, the proposed technique has some theoretical underpinnings for linear-diagonal metrics, but as the authors emphasize that they focus on 'query-access only' metrics, it is not clear whether there are broader theoretical guarantees.
4. How is the reference class selected? The authors claim that the choice has `little impact to the algorithm`, but this is not justified. More specifically, how does the choice of reference class impact the optimization, since the line search is performed based on pairwise comparisons (against the reference class)?

[R1] Snoek, J., Larochelle, H. and Adams, R.P., 2012. Practical bayesian optimization of machine learning algorithms. Advances in neural information processing systems, 25.

[R2] Salimans, T., Ho, J., Chen, X., Sidor, S. and Sutskever, I., 2017. Evolution strategies as a scalable alternative to reinforcement learning. arXiv preprint arXiv:1703.03864.

**Questions:**

**Minor concerns:**
1. Are there any theoretical analyses for the proposed optimization/adaptation procedure if the metric is not linear-diagonal? I realize this might be a big ask, but additional comments on this would be appreciated.
2. Clarifying question: is the validation set/test set sampled from the same target distribution (different from the set the black-box model is trained on)?
3. An important naive baseline to include is directly optimizing $\mathbf{w}$ on some likelihood objective defined over the validation set $\mathcal{S}$. This does not optimize for the metrics of interest $f$, but it is important to see the effects this would have on performance.
4. There is a concern for overfitting or reward hacking of $f$ since $\mathbf{w}$ is tuned solely on (a small) $\mathcal{S}$. The empirical results from the test set seem to indicate this is not a major concern. But can the authors comment on this point with more detail?
5. Importantly, what are the effects of different choices of $\varepsilon$ on post-hoc adaptation performance?

---

### Official Review · Reviewer_ndxP · 2024-10-31

**Soundness:** 2
**Presentation:** 2
**Contribution:** 2
**Rating:** 5
**Confidence:** 4

**Summary:**

The paper proposes CWPlugin, a post-hoc algorithm to optimize a black-box ML model’s performance on a specifci metric. The algorithm iterates over class pairs and optimizes the decision boundaries à la Bayes optimality. The authors proves that the method is theoretically more efficient than brute force search with a binary search implementation in the average case. Empirically, the method is compared against several post-hoc and calibration baselines on prediction tasks.

**Strengths:**

1. CWPlugin is a simple and explainable optimization method, without relying on black-box procedures. Non-ML experts could adopt this algorithm.
2. The algorithm in theory recovers the Bayes optimal classifier.
3. Empirical performance is consistent across datasets, and equally importantly, across varying validation set sizes. It works on several metrics related to the confusion matrix and under distribution shift settings.
4. Relvant to LLM practitioners, the method beats fine-tuning on BERT when data is limited.

**Weaknesses:**

1. Novelty is limited in the case of CWPlugin. The method’s core is finding an optimal vector w to reweight a black-box model’s output. The novel contribution within the algorithm is the local search procedure over pairs of classes. The improvement here is rather incremental. As a more subtle point, the baselines used in Section 4 is inadequantely discussed in the related work subsection. Only probing is mentioned earlier in the paper. This results in a weak qualitative comparison with prior work.
2. A major section of the paper shows CWPlugin’s superior efficiency. This advantage is not demonstrated in the empirical section: there are no concrete runtime comparisons. Speedup from parallelization is also unverified.
3. The presentation of empirical results is not convincing. The tables adopt a specific validation set size, all different depending on the dataset but lacks justification. Sometimes the tables show 4 metrics, and sometimes only 2. This might indicate cherry-picking. The figures show more comprehensive results, but their presentation is poor. Legends are blocking the lines. Figure and axes titles are not formatted properly, and the color bands adds to the confusion. Based on the color bands, CWPlugin does not have an advantage in performance over the baselines due to overlap.
4. Further justification of why the datasets are chosen is necessary. Most of them also require proper citations. More importantly, the authors need to show these datasets as standard for previous works in the same field. For example, the income predicion dataset is uncommon in deep learning from a quick search.
Overall, some of these aspects can be fixed and improve the quality of the paper.

**Questions:**

1. The algorithm’s description is confusing. The algorithm is only iterating over k, but the caption shows iteration over class pairs. The symbol m is abused, sometimes indicating number of classes and sometimes a specific class.
2. What metrics are the original models optimizing? This is important for understanding the results.
3. The authros should re-evaluate whether a number should be bolded in each table. When there is overlap in the error intervals, the result is not usually not considered significant.
4. Table captions should be above the table. Figure 5 is a table. Figures and tables should not be combined together as is the case for Figure 3 and Figure 2. Please present them individually.
5. The abstract on openreview is different from that of the paper. Please converge on a single version of the abstract.

---

### Official Review · Reviewer_fAAR · 2024-11-01

**Soundness:** 2
**Presentation:** 3
**Contribution:** 2
**Rating:** 5
**Confidence:** 4

**Summary:**

In this paper, the author propose CWPLUGIN, a post-hoc method that adapts black-box machine learning models to new target distributions and optimizing specific performance metrics without requiring access to the model's internals (e.g. training details, metric optimizes, hyperparameters used, etc.). CWPLUGIN is a post-hoc method as it takes in as input the set of probabilistic multiclass predictions (e.g. softmax outputs) on a target domain and their corresponding true labels, thereby disregarding any feature information. CWPLUGIN optimizes metrics that are simple functions of the confusion matrix and outputs new class weights, one for each class. The method can be made efficient when the data is balanced across classes or when the metric it is optimizing has a specific quasi-concave shape.

**Strengths:**

The method is computationally very efficient as it solely relies on output probabilities of the black-box model.

The authors also propose ways to further save costs when data is balanced across classes by using parallelization or when the metric it is optimizing has a specific quasi-concave shape; these are interesting observations.

The method is also sample efficient, requiring only a few additional samples to optimize the weights w.

The paper is well-written and the presentation is good overall.

**Weaknesses:**

**Limited novelty**: The authors mention the “probing classifier” approach Hiranandani et al. (2021) that solves a, global linear system in order to find the weights which optimize a particular metric. CWPLUGIN is instead local in that it considers only pair-wise comparisons between classes. Is this the only difference between 'probing classifier' Hiranandani et al. (2021) and CWPLUGIN?
It is unclear if this difference constitutes a significant advancement, especially since the probing classifier already performs very well (e.g. in Figure 6, Probing in fact outperforms in terms of MCC and G-mean. Similarly, Figure 2 shows only marginal improvements in F-measure and accuracy compared to other methods, especially the probing classifier).

**Key literature missing**: The paper could benefit from citing some very important studies in the calibration literature that are relevant to the discussion, including:

1)	Zihao Zhao, Eric Wallace, Shi Feng, Dan Klein, and Sameer Singh. Calibrate before use: Improving few-shot performance of language models. In Marina Meila and Tong Zhang, editors, Proceedings of the 38th ICML 2021.

2)	Abbas, M., Zhou, Y., Ram, P., Baracaldo, N., Samulowitz, H., Salonidis, T., and Chen, T. Enhancing in-context learning via linear probe calibration. In Proceedings of The 27th AISTATS 2024.
3)	Zhixiong Han, Yaru Hao, Li Dong, Yutao Sun, and Furu Wei. Prototypical calibration for few-shot learning of language models. ICLR 2023.

4)	Han Zhou, Xingchen Wan, Lev Proleev, Diana Mincu, Jilin Chen, Katherine Heller, and Subhrajit Roy. Batch calibration: Rethinking calibration for in-context learning and prompt engineering. ICLR 2024.

5)	Zhongtao Jiang, Yuanzhe Zhang, Cao Liu, Jun Zhao, and Kang Liu. Generative calibration for in-context learning. In Houda Bouamor, Juan Pino, and Kalika Bali (eds.), Findings of EMNLP 2023.

6)	M. Shen, S. Das, K. Greenewald, P. Sattigeri, G. Wornell, and S. Ghosh. Thermometer: Towards universal calibration for large language models. ICML 2024.

These papers focus on calibration for LLMs and could also serve as valuable baselines, especially because the authors do studies on language tasks.  Including them could strengthen the paper's comparative analysis and contextualize its contributions within the broader field of language model calibration.

Notably, Zhao et al. (2021) and Abbas et al. (2024) are especially pertinent, since these are also very simple post-hoc methods that use a handful of samples in the range comparable to the one used by the authors (i.e. on the order of tens or hundreds of examples).  While these papers don't directly address the specific metrics used by the authors, they demonstrate high utility in terms of accuracy. It would be valuable to evaluate their performance on the metrics employed in this study, such as F-measure, MCC, and G-mean. Comparing CWPLUGIN against these methods would provide a more comprehensive assessment of its effectiveness.


**Marginal improvement over baselines**: Figure 2 shows only marginal improvements in F-measure (e.g. 0.579 vs 0.576) and accuracy (e.g. 0.619 vs 0.617) compared to other methods, especially the probing classifier. Moreover, in Figure 6, Probing in fact clearly outperforms CWPLUGIN in terms of MCC and G-mean.

Similarly, in Figure 3, on Language tasks, we observe marginal improvement over FullDirich baseline on most metrics except F-measure (e.g. 0.563 vs 0.562 on accuracy, 0.256 vs 0.255 on MCC).

Compared to the baselines, CWPLUGIN is the least performing method in terms of accuracy and MCC for lmemotions (figure 8) and one of the least performing methods on lmemotionsOOD (Figure 9). The results are perhaps good for applications where different evaluation metrics (e.g. F-measure) are more critical than mere predictive accuracy. However, I think even improvements on these metrics are marginal in most cases.

**Figure 5 table ANLI with label noise results for G-mean do not match results in Figure 13 in the appendix. In Figure 13, Clean baseline always outperforms CWPLUGIN for all Validation Set sizes where as in Figure 5, Clean has lower mean of 0.528 as compared to 0.541 of CWPLUGIN.**

**Reproducibility**: The code for reproducing the results has not been provided, which may raise concerns about the reproducibility of the findings. Making the code available would enhance transparency.

**Minor comments**:
Line 263: typo: “…linear-diagonal metric Both results…” has a missing full-stop.

**Questions:**

**Clarification regarding Section 4.1 setting**: In Section 4.1, could you elaborate on how the regression model is used for classification tasks? What is the process for converting regression outputs into discrete class labels? Are there predefined thresholds or ranges used to map continuous outputs to specific classes?

**Clarification regarding section 4.2.1 setting**: distilBERTbased is used a the base model while a finetuned version of BERT is used as a baseline. What is the reason behind choosing a different model as a basline? Would it be more appropriate to use a fine-tuned version of distilBERT (distilBERT-FT) as the baseline for a more direct comparison? Alternatively, why not use BERT as the base black-box model to maintain consistency with the baseline (perhaps a more apples-to-apples comparison)? How might these different choices impact the interpretation of the results and the conclusions drawn from the comparison?

**Clarification regarding section 4.2.2 setting**: Why test with different base models for lmemotions and lmemotionsOOD.? how about using the same base model across all tasks? If different base models are necessary, could an additional ablation study be conducted using a single base model across all tasks to isolate the effect of the model choice?

**Clarification regarding section 4.3 setting**: The authors mention on lines 484-485 that ‘For both the label shift and label noise settings, we utilize a model trained on GLUE (Wang et al., 2019) and ANLI as our base, black-box predictor’ but did not mention which model used. What model/LLM is used for this task?

---

### Official Review · Reviewer_namq · 2024-11-04

**Soundness:** 2
**Presentation:** 3
**Contribution:** 2
**Rating:** 3
**Confidence:** 5

**Summary:**

Black-box or proprietary models often limit users to accessing only predictions via API calls, making it difficult to adapt the model behaviers to align with specific user preferences. This paper introduces CWPLUGIN, a plug-and-play method that reweights prediction probabilities to match users' desired metrics or target domains. CWPLUGIN uses a coordinate-wise search algorithm that iteratively finds optimal relative weights for each class pair to achieve the desired reweighting. Empirical results show that CWPLUGIN outperforms existing calibration methods and fine-tuning approaches, especially when data size is limited. Additionally, CWPLUGIN proves effective in handling scenarios with label shift and label noise. This framework offers new opportunities for users to adapt black-box models to their tasks, allowing for a degree of customization previously unavailable.

**Strengths:**

**Originality:** While CWPLUGIN shares the concept of post-processing prediction probabilities through reweighting, it is uniquely motivated by the need to adapt black-box models to specific target domains, providing a fresh perspective on model adaptation for restricted-access models.

**Quality:** CWPLUGIN demonstrates strong performance across a range of scenarios without requiring large amounts of validation data. And its design can be adapted to many settings. Their experiment shows promising results.

**Clarity:** The motivation for CWPLUGIN is clear to me. However, the algorithmic section could benefit from further clarification (see below).

**Significance:** This paper addresses a critical and emerging challenge in the field: adapting black-box models for task-specific needs. The proposed method shows effectiveness and efficiency by adapting predictions through simple probability reweighting, providing a practical solution for the community’s growing demand for adaptable AI tools.

**Weaknesses:**

**Algorithm 1:** The CWPLUGIN algorithm iterates over each class pair; however, the notation used is unclear. The symbol m denotes the number of classes but is also used as a class index, which creates confusion. Additionally, in line 3 of the algorithm, a single for-loop is shown, which does not accurately reflect the process of iterating over class pairs. Using a nested for-loop structure would better clarify this.

**Class Size:** The experiments were conducted on datasets with a relatively small number of classes. It would be beneficial to include results on datasets with a larger number of classes, such as adapting to specific metrics on CIFAR-100, TinyImageNet, ImageNet, to demonstrate the scalability of CWPLUGIN when m is high.

**API Call Expense for Validation:** The paper could discuss how the cost of API calls increases with the size of the validation dataset, which may be a limitation in scenarios where extensive validation data is needed to reach desired performance.

**Suggested References:** Two relevant papers are suggested to read and include:
* The first paper leverages relational information in label space to reweight prediction probabilities without requiring validation datasets: https://arxiv.org/abs/2307.12226.
* The second paper adapts black-box models by steering model outputs using a combination of tuned and untuned models: https://arxiv.org/abs/2401.08565.

**Questions:**

1. Does the order of class pairs impact the iterative process for finding optimal weights?
2. CWPLUGIN currently does not take class frequency or class imbalance in the validation dataset into account. Could incorporating these factors enhance performance, particularly in label-shift scenarios?
3. Why was BERT-FT chosen as a baseline for comparison instead of fine-tuning other used black-box models like DistilBERT, RoBERTa, or DistilRoBERTa? For example, section 4.2.1, black-box model is DistilBERT, but we are comparing to BERT-FT.

---

### Author Response · Authors · 2024-11-27
**Withdrawing submission**

We thank all the reviewers for their valuable feedback, and have decided to incorporate the changes in a future version of the paper. Thank you so much for your time reviewing, and we hope that we can improve the paper to meet your expectations in the future!

---

### Note · Authors · 2024-11-27

**Comment:**

We thank all the reviewers for their valuable feedback, and have decided to incorporate the changes in a future version of the paper. Thank you so much for your time reviewing, and we hope that we can improve the paper to meet your expectations in the future!

**Withdrawal Confirmation:**

I have read and agree with the venue's withdrawal policy on behalf of myself and my co-authors.